# Highly sensitive spatial transcriptomics using FISHnCHIPs of multiple co-expressed genes

Xinrui Zhou [1,2], Wan Yi Seow [1,2], Norbert Ha [1], Teh How Cheng[1],
Lingfan Jiang[1], Jeeranan Boonruangkan[1], Jolene Jie Lin Goh [1],
Shyam Prabhakar [1], Nigel Chou [1] ✉ & Kok Hao Chen [1] ✉

High-dimensional, spatially resolved analysis of intact tissue samples promises to transform biomedical research and diagnostics, but existing spatial omics technologies are costly and labor-intensive. We present Fluorescence In Situ Hybridization of Cellular HeterogeneIty and gene expression Programs (FISHnCHIPs) for highly sensitive in situ profiling of cell types and gene expression programs. FISHnCHIPs achieves this by simultaneously imaging ~2-35 co-expressed genes (clustered into modules) that are spatially co-localized in tissues, resulting in similar spatial information as single-gene Fluorescence In Situ Hybridization (FISH), but with ~2-20-fold higher sensitivity. Using FISHnCHIPs, we image up to 53 modules from the mouse kidney and mouse brain, and demonstrate high-speed, large field-of-view profiling of a whole tissue section. FISHnCHIPs also reveals spatially restricted localizations of cancer-associated fibroblasts in a human colorectal cancer biopsy. Overall, FISHnCHIPs enables fast, robust, and scalable cell typing of tissues with normal physiology or undergoing pathogenesis.

Enormous cellular diversity arises when a single cell develops into an organism. Recent advancements in single-cell RNA-sequencing (scRNA-seq) make it possible to unbiasedly define cell types reflecting ontogeny, functions, or anatomical locations. However, high-throughput mapping of these cells within intact biological systems is still a technical challenge[1-3]. Technologies that enable highly multiplexed mapping of cell types within the context of normal tissue physiology or tumor microenvironment could provide valuable insights into the diverse biological processes that contribute to human health and disease.

Spatial indexing combined with next-generation sequencing has enabled spatial mapping of sequencing reads and in situ reconstructions of cell types[4-6], but sequencing-based spatial transcriptomics methods are limited by RNA diffusion and capture efficiency. Cell types can also be imaged by targeting RNAs with multiplexed single-molecule FISH (smFISH) or in situ sequencing[7-14]. Imaging-based

spatial transcriptomics methods are highly quantitative and scalable to the whole transcriptome (~10,000 genes)[15,16], but they require high-resolution microscopes, become more laborious with more targets or need prioritization of marker genes[17-20], may suffer from non-specific noise[21], and are limited by molecular crowding[13,22,23]. Another approach is multiplexed immunostaining or spatial proteomics[24-26]. While the increased copy number of proteins compared to RNAs may lead to more robust detection, antibody panels are costlier, less flexible, and the assays have lower throughputs.

Here, we introduce a sensitive, robust, and scalable FISH-based spatial transcriptomics method that profiles single cells using multiple co-expressed genes. We reason that since co-expressed genes are spatially co-localized in the same cells within a tissue, one can design RNA FISH probes to target a large set of genes to detect any cell population of interest more reliably (Fig. 1a). We call the method Fluorescence In Situ Hybridization of Cellular HeterogeneIty and gene

[1]Genome Institute of Singapore, Agency for Science, Technology and Research (A*STAR), 60 Biopolis Street, Singapore 138672, Singapore. [2]These authors contributed equally: Xinrui Zhou, Wan Yi Seow. ✉e-mail: nigel_chou@gis.a-star.edu.sg; chenkh@gis.a-star.edu.sg

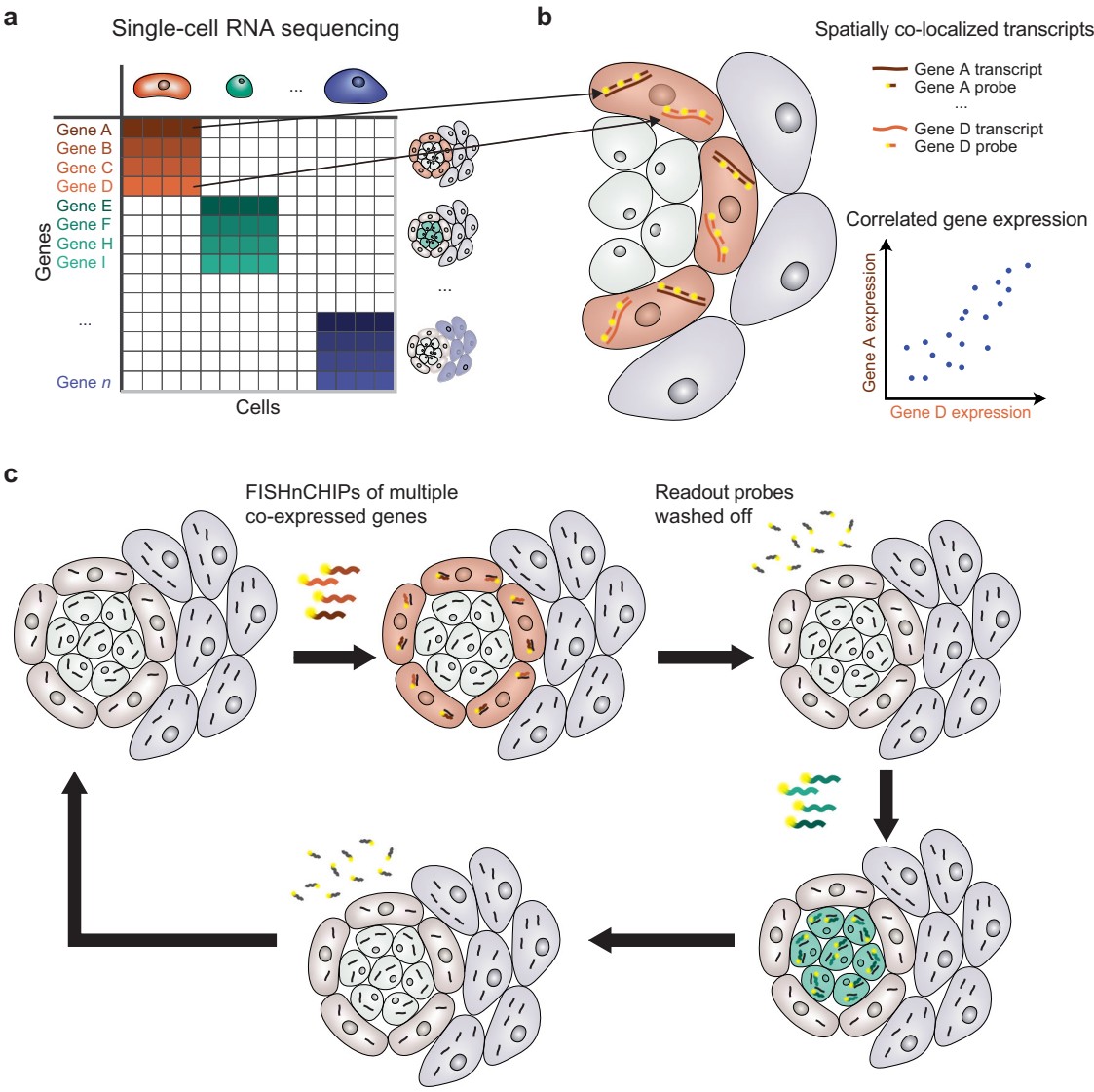

**Fig. 1 | FISHnCHIPs schematic. a** Cell-by-gene count matrix from single-cell RNA sequencing (scRNA-seq) can be used to cluster cell types, which are characterized by their unique gene expression profiles (red cells express genes A–D; green cells express genes E–I). **b** Genes that are co-expressed with each other are spatially co-localized in the same cells within a tissue. By designing fluorescently labeled oligonucleotide probes to target a large set of co-expressed transcripts, FISH-nCHIPs can improve the sensitivity of fluorescence detection. **c** Combined with repeated rounds of hybridization and washing, FISHnCHIPs enables robust and scalable mapping of cell types in tissue samples.

expression Programs (FISHnCHIPs) and show that it can accurately map cell types while preserving tissue architecture. Using a reference scRNA-seq dataset, we identify groups of correlated genes and design thousands of oligonucleotide probes against their transcripts, which result in tens of thousands of fluorescent tags per cell (factoring in number of genes, transcript copy number per cell, and number of probes per transcript) (Fig. 1b). By further taking advantage of the array-synthesized oligo-pool and sequential fluidics technologies (Fig. 1c), we seek to demonstrate FISHnCHIPs based expression profiling in mouse kidney, mouse brain, and human colorectal cancer tissues.

## Results

For the initial testing of our hypothesis, we used a mouse kidney scRNA-seq dataset[27] to design FISHnCHIPs probes for five selected cell types: renal macrophages, glomerular endothelial cells, loop of Henle (LOH) cells, collecting duct (CD) cells, and glomerular podocytes—a rare cell type that functions as a renal filtration barrier (Fig. 2a). We

observed a high degree of co-localization between the top two co-expressed genes in each of these cell types, confirming that correlated genes from scRNA-seq are indeed spatially co-localized in the same cells (Fig. 2b). When using a combination of 14–23 genes to label each of these cell types (Fig. 2c and Supplementary Data 1), the cells were much more easily detected compared to labeling only the single top differentially expressed (DE) gene. Although these 5 cell types represent only ~12% of the total kidney cell population (estimated from scRNA-seq), FISHnCHIPs revealed many intricate details of the kidney tissue architecture, such as the arrangement of podocytes in the highly fenestrated Bowman's capsule, where they wrap around the glomerular endothelial cells (Fig. 2d–i).

The FISHnCHIPs fluorescence intensity per cell was increased by ~6–39 fold across the five cell types (median of at least 146 cells) compared to smFISH (Supplementary Fig. 1a). However, we observed that some of the FISHnCHIPs genes were also expressed in off-target cell types from the scRNA-seq data (Fig. 2a). For example, *Slc5a3*, which has a Pearson's correlation (*r*) of 0.33 to *Slc12a1* (a marker for LOH), is

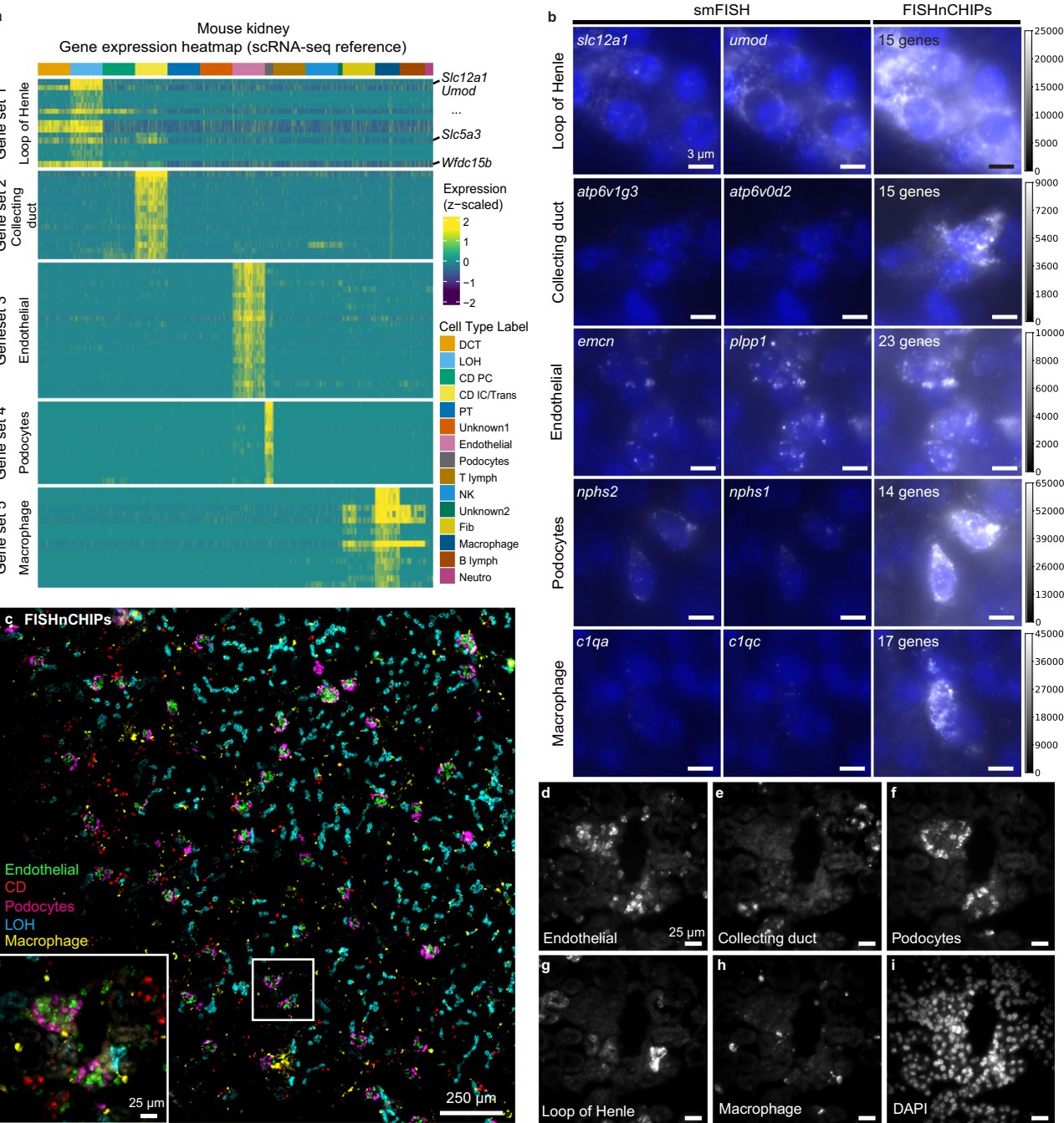

**Fig. 2 | Comparison of FISHnCHIPs and single-molecule RNA FISH (smFISH) in mouse kidney tissue. a** Gene expression heatmap from the scRNA-seq reference data and their corresponding cell clusters. Heatmap shows expression of the 84 FISHnCHIPs genes that are correlated to the top differentially expressed (DE) genes in the 5 selected cell types, sampling a maximum of 300 cells per cluster. Cell type labels are distal convoluted tubule (DCT), loop of Henle (LOH), collecting duct principal cell (CD PC), collecting duct intercalated cell/collecting duct transitional cell (CD IC/Trans), proximal tubule (PT), unknown1, endothelial cell, podocytes, T-lymphocytes (T lymph), natural killer cell (NK), unknown2, fibroblasts (fib), macrophage, B lymphocytes (B lymph), neutrophil (neutro). **b** Unprocessed

smFISH images of a mouse kidney tissue slice in 5 selected cell types are shown in the left and middle panels. FISHnCHIPs labels 14–23 co-expressed genes simultaneously to detect target cell types and their images are shown in the right panels. The smFISH and FISHnCHIPs images are scaled to the same camera intensity range for each cell type. Nuclei staining with DAPI is shown in blue. Scale bar, 3 μm. **c** Composite FISHnCHIPs image with each cell type assigned a pseudo color: endothelial (green), collecting duct (red), podocyte (magenta), loop of Henle (blue) and macrophage (yellow). Scale bar, 250 μm. Insert: Zoomed-in region of the white box. Scale bar, 25 μm. **d–i** Zoomed-in region of the white box insert in (**c**). Scale bar, 25 μm.

also expressed in CD cells. To estimate the crosstalk in FISHnCHIPs data, we computed the Manders' overlap coefficient across the five cell-type channels, which ranged from 0.001 to 0.09, suggesting minimal crosstalk for these cell types (Supplementary Fig. 1b). To computationally predict the trade-off between improved sensitivity and potential misidentification for all cell types, we leveraged the

scRNA-seq reference data to compute two metrics, Signal Gain (SG) and Signal Specificity Ratio (SSR), both expressed as a function of the number of genes used (ranked by their Pearson's correlation to the top DE gene) (Supplementary Fig. 2). We defined SG as the ratio of the sum of counts for FISHnCHIPs genes to that of the top DE gene, and SSR as the ratio of the sum of counts for FISHnCHIPs genes in the target cell

type to that in the most likely off-target cell type. When SSR approaches unity, the fluorescence intensity for the cell type of interest should be equal to that of an off-target cell type, rendering them indistinguishable. Disappointingly, we found that 7 out of the 16 previously annotated kidney cell types have a SSR of less than 4, reflecting a lack of specificity for these cells when using this cell-centric design and motivating the development of a more refined approach.

We asked if there is a way to design FISHnCHIPs gene-sets that naturally diminishes crosstalk. We reasoned that since metazoan genomes are organized by pathways and regulatory modules that exhibit coordinated expression variability[28], the imaging of gene modules (groups of correlated genes) should result in spatially

coherent FISHnCHIPs signals. Motivated by this idea, we sought to demonstrate a gene module-based FISHnCHIPs assay (that is without the a priori clustering of cell types). We performed clustering of the gene-gene correlation matrix (instead of the gene cell matrix) of a mouse visual cortex dataset[29], selected 255 candidate genes which are highly correlated (Pearson's correlation ($r$) > 0.7) to at least three genes, and identified 18 gene modules with significant enrichment for Gene Ontology (GO) (Fig. 3a and Supplementary Data 1–3). By profiling gene modules (an average of 14 genes per module), FISHnCHIPs signals were predicted to be 1.2 to 22.3-fold brighter than profiling with individual marker genes. Furthermore, in this gene-centric design, signals from different modules have low crosstalk with each other

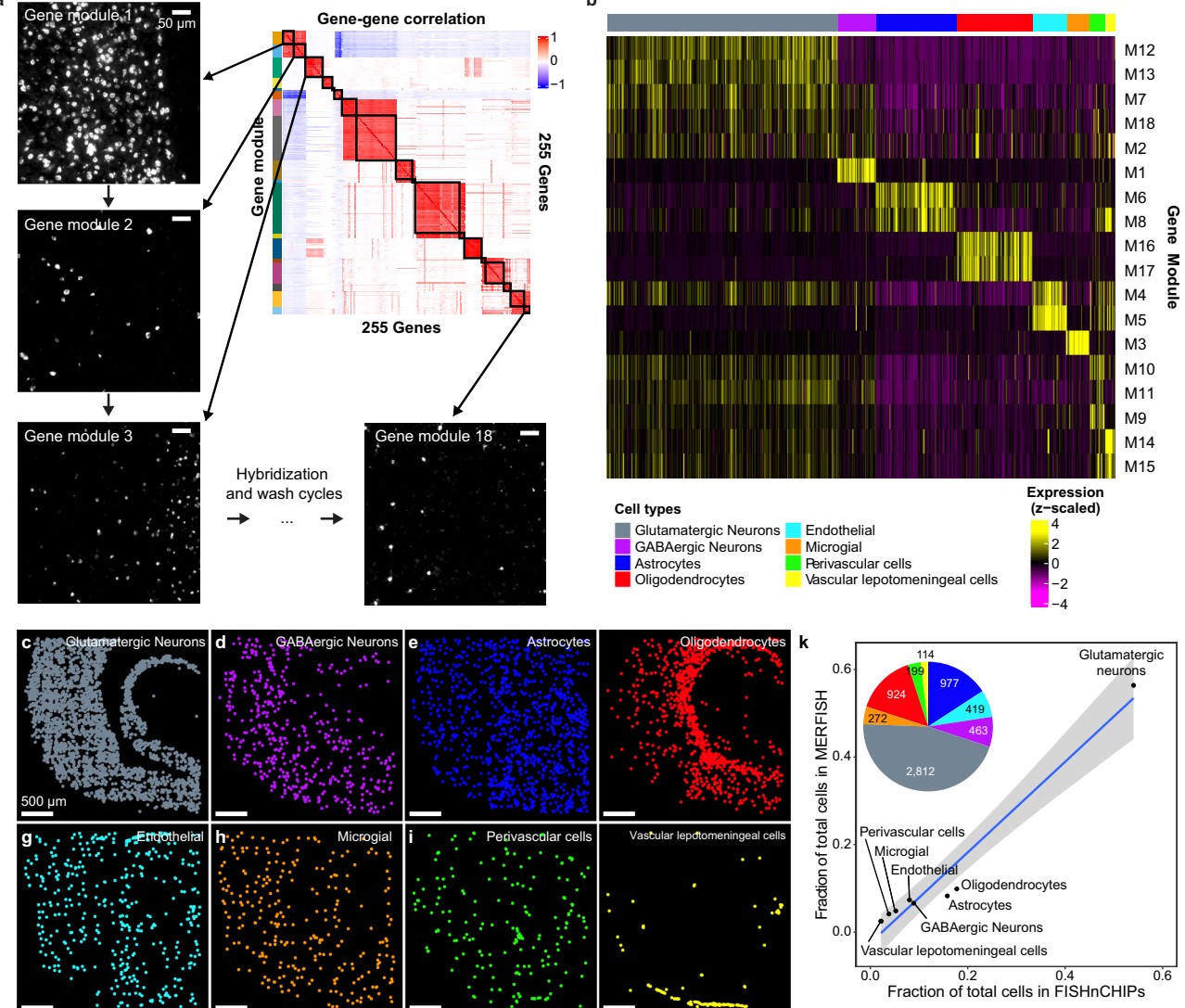

**Fig. 3 | FISHnCHIPs profiling of 18 gene modules in the mouse cortex. a** Gene-gene correlation heatmap (of the pairwise Pearson's correlation ($r$) coefficients) grouped into 18 clusters of gene modules (colored boxes, M1-M18). Each module (14 genes on average) was imaged sequentially under an automated fluidics-coupled fluorescence microscope system. Example FISHnCHIPs images of a mouse brain tissue slice stained for gene module 1, 2, 3, and 18. Scale bar, 50 μm. **b** Single cells were segmented using DAPI stain and the cell masks were applied to define 6180 cells after quality control. Heatmap of the mean fluorescence intensity per cell for each imaged module. The cell-by-module intensity matrix was clustered using the Louvain algorithm, resulting in eight cell clusters. **c**–**j** Spatial maps of the

detected cells, colored by their cell types: Glutamatergic neurons (gray), GABAergic neurons (purple), astrocytes (blue), oligodendrocytes (red), endothelial cells (cyan), microglial cells (orange), peri-vascular cells (green), and vascular leptomeningeal cells (yellow). Scale bar, 500 μm. **k** Scatter plot of cell type frequency detected by MERFISH versus FISHnCHIPs fitted with a linear regression model: $y = -0.025 + x, R^2 = 0.96$, where $x, y$ is the cell type frequency in FISHnCHIPs and in MERFISH, respectively. The gray band surrounding the regression line represents the 95% confidence interval for the linear regression model. Insert is a pie chart showing the proportion of each FISHnCHIPs cluster. Source data are provided as a Source Data file.

(Supplementary Fig. 3). We imaged each gene module sequentially (Fig. 3a) in a fresh frozen mouse brain tissue section using an automated fluidics system, and created a software pipeline to align, segment, and cluster cell types based on the FISHnCHIPs data (Fig. 3b, Supplementary Fig. 4). Unsupervised graph-based clustering of the gene module expression from 6180 cells recapitulated the expected frequency and spatial distribution of the major brain cell types, including glutamatergic neurons, GABAergic neurons, astrocytes, oligodendrocytes, endothelial, microglial, peri-vascular, and vascular leptomeningeal cells (Fig. 3c–j). The frequency of cell types detected by FISHnCHIPs was highly correlated to MERFISH[30] (Pearson's correlation $r = 0.98$) (Fig. 3k and Supplementary Data 4) and was reliably detected across technical replicates (Supplementary Fig. 5).

Next, we showed that FISHnCHIPs can be used to distinguish the neuronal subtypes that stratify the canonical laminar structure of the visual cortex. A dimensionality reduction-based algorithm—consensus non-negative matrix factorization (NMF)[31], was previously shown to infer coordinated gene expression in neurons. So, we performed gene-gene correlation analysis of the 20 previously annotated identity and activity gene expression programs and designed a FISHnCHIPs panel containing an average of 16 genes per program with SG ranging from 1.2 to 7.6 (Supplementary Data 1–2, and Supplementary Fig. 6). The neuronal programs were clearly detected by FISHnCHIPs: the 14 identity programs (ExcL2, ExcL3… Sub) were more spatially localized while the 6 activity programs (Erp, LrpD… Syn) were more ubiquitously expressed (Fig. 4a–f). Clustering analysis of 2794 cells with the identity programs revealed 11 neuronal clusters (Fig. 4g–s). We quantified the distribution of excitatory and inhibitory neurons along the cortical depth and observed that the excitatory neurons were organized into six layers (Fig. 4t). The inhibitory neurons also displayed layer-specific localizations, with IntNpy/CckVip being more concentrated in the upper layers, whereas the IntSst and IntPv neurons populated the deep layers (Fig. 4u–v), consistent with previous findings[30,32]. Importantly, FISHnCHIPs also revealed subtle spatial variations of these gene expression programs. We found that the excitatory programs (except for ExcL6p1) varied continuously with distance to the outer edge of the cortex, and some programs had expression distributions that partially overlapped along the cortical depth (Supplementary Fig. 7), suggesting that spatial gene expression gradients could underlie the continuous nature of neuronal sub-types[30,33].

Having established the reliability of FISHnCHIPs for profiling modules and programs, we sought to demonstrate how it improves imaging throughput over conventional single gene-based methods. With the improved signal, cells labeled with FISHnCHIPs were well detected even when imaged under low magnification, thus enabling larger fields of view and more cells to be profiled in the same amount of time (Supplementary Fig. 8). In addition, we sought to demonstrate a more comprehensive cell typing for both neuronal and non-neuronal cells within the mouse brain. We turned to an unsorted scRNA-seq dataset[34] and designed a FISHnCHIPs assay containing 53 modules (674 genes), with SG ranging from 1.9 to 20.2 (Supplementary Fig. 9a, and Supplementary Data 1–3). To further evaluate the panel, we simulated the 53 FISHnCHIPs modules with the scRNA-seq dataset (sum of gene expression within each module) and assessed the clustering accuracy with respect to the reference annotations using the Adjusted Rand Index (ARI) (Supplementary Fig. 9b–f). The 53 modules panel has an ARI score of 0.814, suggesting that it could recapitulate the known brain cell types to a large extent. For comparison, the ARI score with 1000 highly variable genes (simulating a conventional assay profiling 1000 genes individually) is only slightly higher at 0.846.

We proceeded to image a whole tissue section using the 10× instead of 60× objective lens (see Supplementary Fig. 10 for example images). This allowed us to cover a 36-fold larger area in the same amount of assay time (21 h). We observed that co-expressed gene modules co-localized in the same cells and biologically related

modules clustered closely in the expression space (Fig. 5a). Unbiased clustering of 54,834 cells revealed 18 cell types with unique gene expression profiles and spatial organization, including inhibitory and excitatory neurons, astrocytes, microglia, pericytes, oligodendrocytes, ependymal cells, endothelial cells, and other blood vessel associated cells (Fig. 5b–u). Cell types identified by FISHnCHIPs showed good correspondence to scRNA-seq (Supplementary Fig. 11). Further sub-clustering of some of the cell types uncovered finer subtypes with distinctive spatial distributions (Supplementary Fig. 12). For example, we observed distinct localizations for the subtypes of blood vessel associated cells, such as *CNN1*+ smooth muscle cells, *DCN*+ fibroblasts[35], *MRC1*+ (also known as *CD206*) border-associated macrophages[36] that resided almost exclusively at the cortical surface, and *GKN3*+ arterial endothelial cells that formed large penetrating vascular structures. To further validate the performance of our high throughput FISHnCHIPs assay, we compared the frequency and spatial distribution of cell types observed under 10× versus 60× objectives using two closely adjacent cryo-sections. The cluster sizes were highly correlated between the 10× and 60× datasets (Pearson's correlation, $r = 0.95$), indicating that there was no observable degradation of data quality despite the increased throughput (Supplementary Fig. 13). Notably, we observed that a much higher proportion of cells (97%) passed quality control (QC) in this 53-module library compared to previous libraries (Supplementary Fig. 14). This indicates that if FISHnCHIPs is designed to target a broad range of cell types, almost all cells imaged can be profiled, further demonstrating the high detection efficiency and sensitivity of FISHnCHIPs.

Another advantage of FISHnCHIPs is its robustness. The use of redundant genes and high SG should facilitate the imaging of clinical samples that may suffer from lower RNA quality. As a final demonstration, we used FISHnCHIPs to analyze a frozen biopsy of human colorectal cancer (CRC) tissue by imaging two cancer-associated fibroblasts (CAFs) subtypes that we previously identified from a scRNA-seq study[37] (Fig. 6a–f and Supplementary Fig. 15). We also co-stained the epithelial cells (tumor marker genes) and immune cells (human leukocyte antigen, HLA genes) in the CRC tissue using FISHnCHIPs. We observed distinct spatial organization of the two CAF subtypes, with the CAF-2 subtype expressing the muscle contraction program appearing to promote an immuno-suppressive micro-environment, where fewer immune cells (0.74-fold, $p = 1.4 \times 10^{-72}$ (2-sided Mann-Whitney U test) were detected in the vicinity of CAF-2 compared to CAF-1 (Fig. 6g–h and Supplementary Fig. 16). The observed staining pattern is in agreement with immunofluorescence (IF) labeling (Supplementary Fig. 17). In contrast, smFISH against individual genes, *DCN* or *MMP2* (markers for CAF-1), as well as *TAGLN* or *ACTA2* (markers for CAF-2), were dim and the CAFs subtypes were hardly distinguishable from the hazy background fluorescence (Supplementary Fig. 18), reinforcing the benefit of labeling cell types with multiple co-expressed genes.

In summary, we showed that FISHnCHIPs can be used to robustly image cell types in mouse and human tissues, as well as to characterize cell types through unsupervised clustering. scRNA-seq studies have repeatedly highlighted the importance of gene modules or transcriptional programs in diseases, but they have not been imaged directly. We leveraged the key insight that correlated genes are spatially coherent to image them via in situ hybridization in a high throughout manner. We found that some gene modules, or combinations of modules, marked discrete cell populations localized to specific brain regions while others varied continuously along certain spatial axes in the brain. Accurate mapping of both types of transcriptional state is critical for understanding the phenotypic properties of any tissue.

## Discussion

The key advantage of FISHnCHIPs is that it provides ~2–20-fold higher sensitivity for mapping cell types (depending on the desired

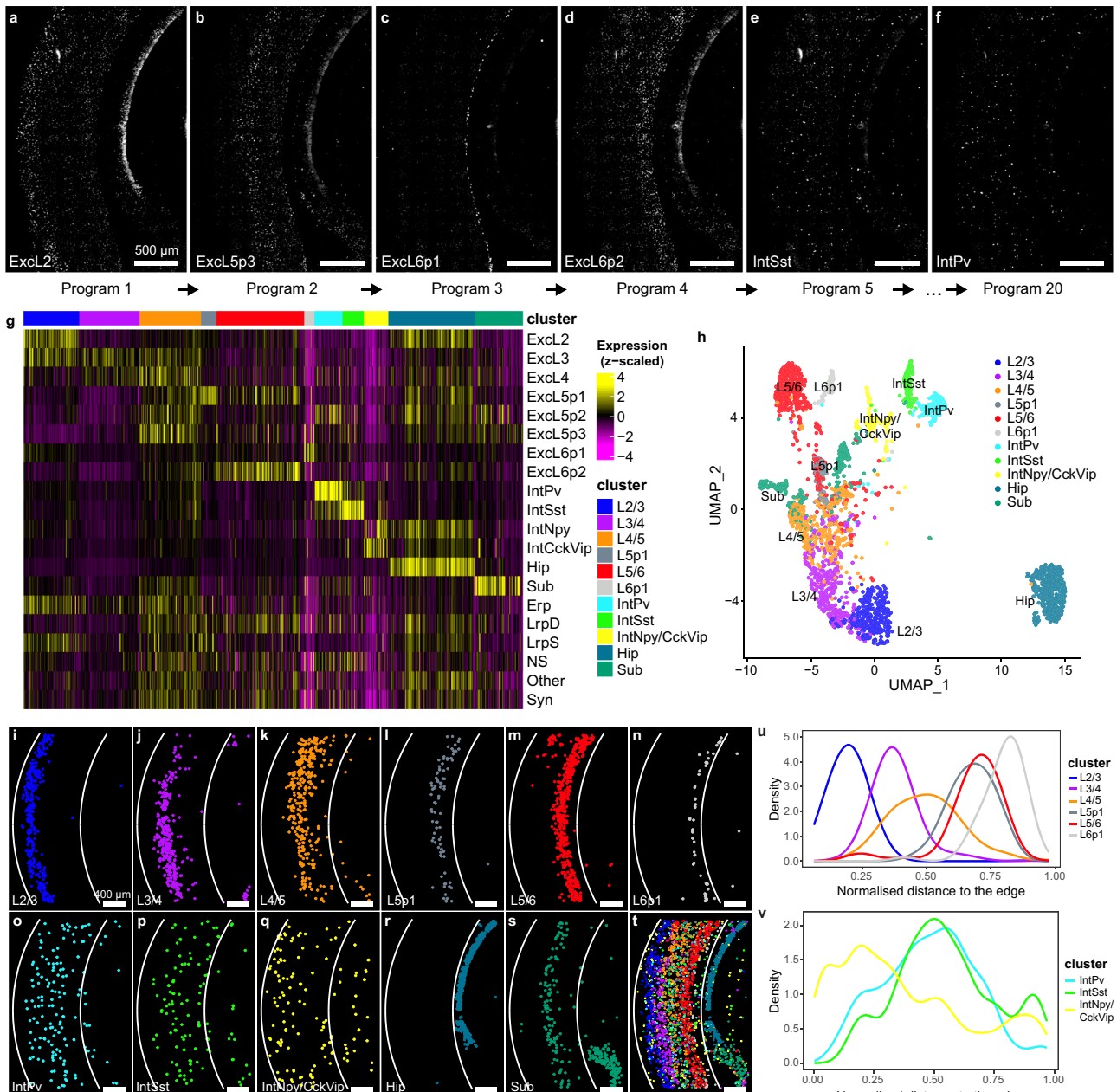

**Fig. 4 | FISHnCHIPs profiling of 20 gene expression programs in the mouse cortex. a–f** 6 out of 20 FISHnCHIPs images of a mouse brain tissue slice stained for programs ExcL2, ExcL5p3, ExcL6p1, ExcL6p2, IntSst, and IntPv. An average of 16 co-expressed genes were imaged concurrently. Scale bar, 500 μm. **g** Single cells were segmented and a total of 2794 passed quality control. Heatmap of the mean fluorescence intensity per cell for each imaged program. The cell-by-program intensity matrix was clustered using the Louvain algorithm, resulting in 11 clusters. We used the program annotations to label the cluster identities. **h** Uniform manifold approximation and projection (UMAP) colored by cluster. **i–s** Spatial maps of the detected cells, colored by their cell types: L2/3 excitatory neurons (blue), L3/4 excitatory neurons (purple), L4/5 excitatory neurons (orange), L5p1 excitatory

neurons (gray), L5/6 excitatory neurons (red), L6p1 excitatory neurons (white), IntPv inhibitory neurons (cyan), IntSst inhibitory neurons (light green), IntNpy/CckVip inhibitory neurons (yellow), hippocampus (light blue), and subiculum (dark green). Scale bar, 400 μm. **t** Composite image of the detected cell clusters. Cortical depth distance (for **u**, **v**) was calculated based on the two white arcs (see materials and methods). Some programs exhibited gradual intensity variation along the cortical depth (see cell-by-program intensity heatmap ordered by increasing cortical depth in Supplementary Fig. 7). Kernel Density Estimate (KDE) of cell density for excitatory neurons (**u**) and inhibitory neurons (**v**) along the cortical depth. Source data are provided as a Source Data file.

cell type "resolution") compared to conventional FISH. In contrast to existing marker genes selection strategies that minimize redundancy[18,38] or use compressed sensing to improve the multiplexing efficiency for individual genes[17], FISHnCHIPs leverages the redundancy of correlated genes to boost sensitivity and robustness. For example, the amplified signal can be harnessed for rapid whole-tissue imaging under a lower magnification objective lens. The high-throughput, large field-of-view profiling enabled by FISHnCHIPs

could facilitate detection of rare cell populations within complex tissues. For instance, stem cell niches housing slowly dividing stem cells often occur at low frequencies interspersed throughout a tissue[39]. By performing rapid whole-tissue scans, FISHnCHIPs can help identify these niche locations based on their expression signatures. Linking rare cell gene expression patterns to their surrounding microenvironments could resolve external events or cell-cell interactions that influence cell fates. This is especially important

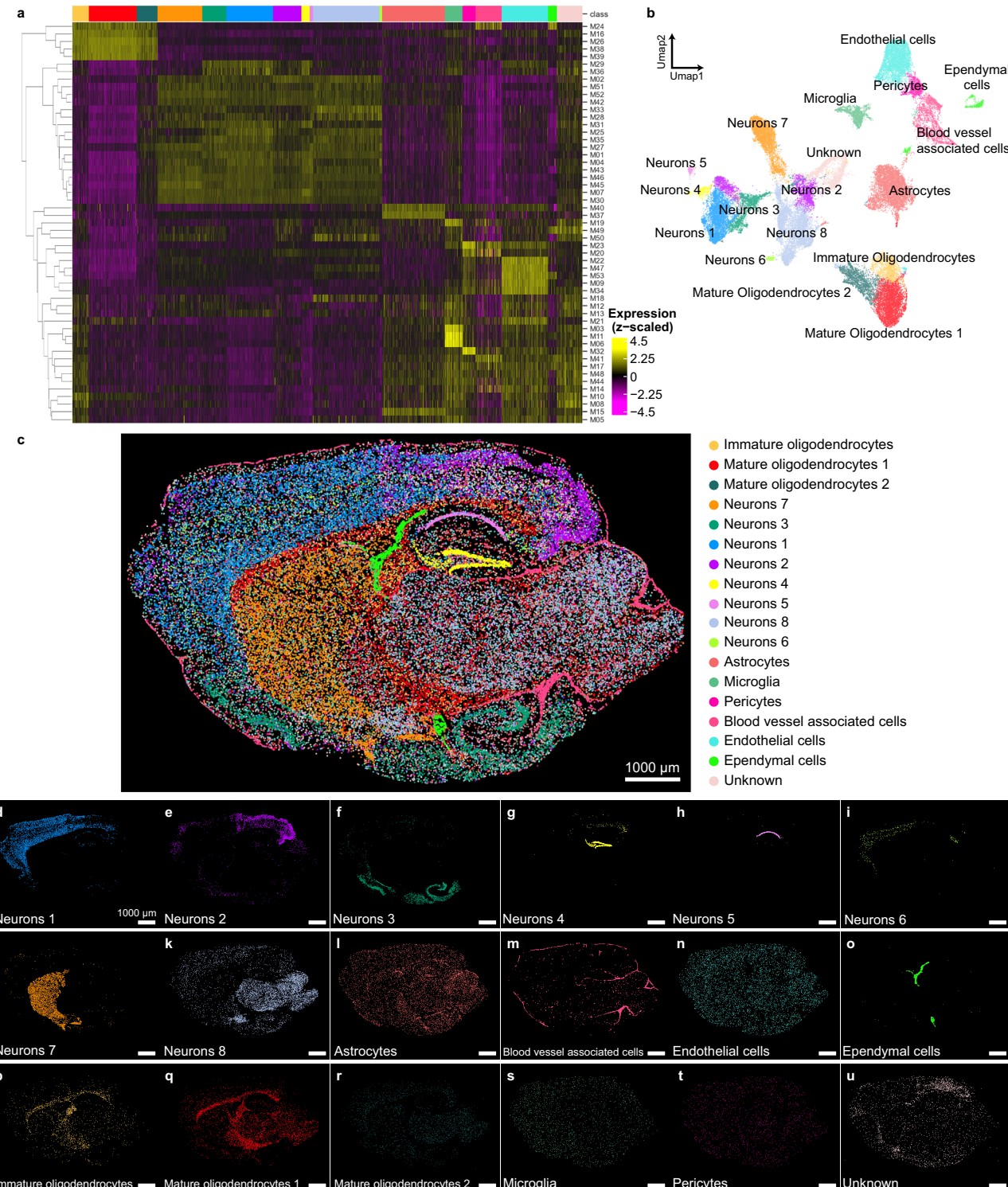

**Fig. 5 | Large FOV FISHnCHIPs profiling of 53 gene modules in the mouse brain.** **a** The cell-by-module intensity matrix of the FISHnCHIPs data for 54,834 cells was clustered to reveal 18 major cell types. Sub-clustering revealed finer subtypes that were also spatially distinct (see Supplementary Fig. 12). **b** Uniform manifold approximation and projection (UMAP) representation for all cells, colored by their cell types. **c** Composite image of the detected cells, colored by their cell types. Scale bar, 1000 µm. **d–u** Spatial map of the 18 clusters: neurons 1, 2, 3, 4, 5, 6, 7, and 8, astrocytes, blood vessel associated cells, endothelial cells, ependymal cells, immature oligodendrocytes, mature oligodendrocytes 1 and 2, microglial, pericytes, and unknown. Scale bar, 1000 µm. Color scheme is the same for (**a–u**). Source data are provided as a Source Data file.

for revealing the potentially pathologic roles of certain tumor microenvironments in seeding cancer stem cells.

By utilizing redundant genes, FISHnCHIPs is also more robust when analyzing clinical tissues. Furthermore, optical crowding in small cells would typically hinder the accurate decoding of highly expressed RNA transcripts, but FISHnCHIPs turns this into an advantage by simultaneously profiling co-localized genes at the level of single cells. FISHnCHIPs may be combined with split-probe[21], tissue clearing[40], or

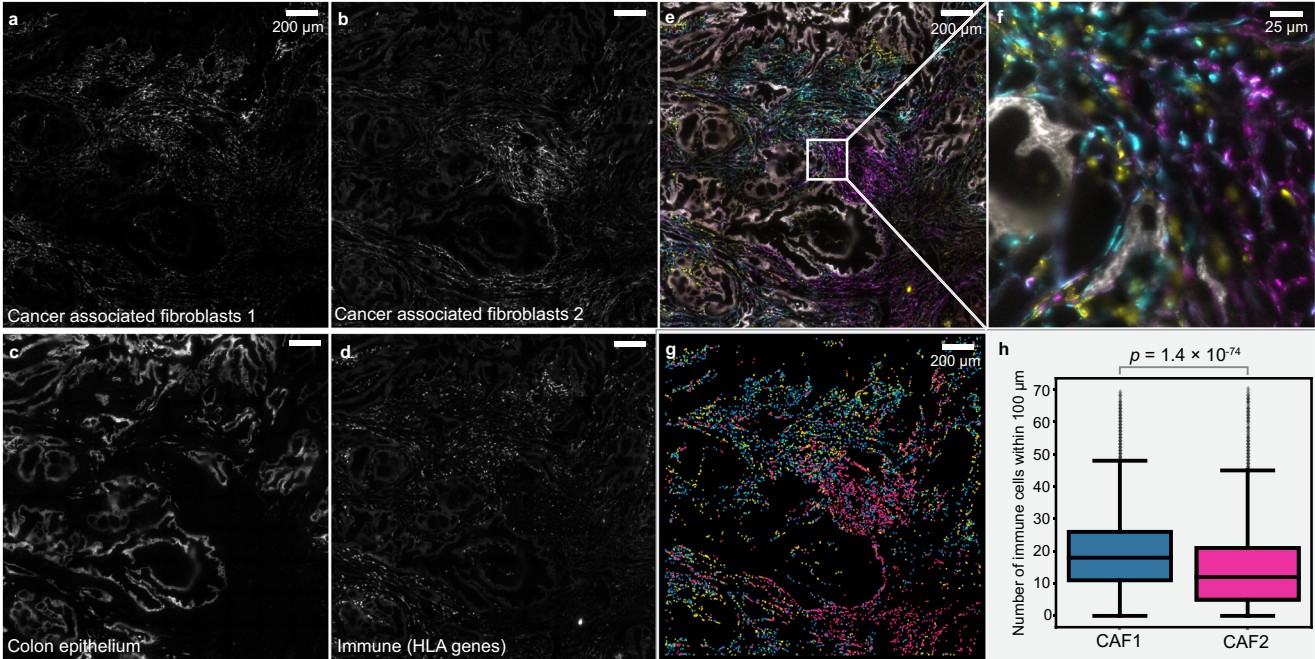

**Fig. 6 | FISHnCHIPs imaging of cancer associated fibroblasts (CAFs) subtypes in human colorectal cancer (CRC) tissue. a–d** FISHnCHIPs images of cancer associated fibroblasts 1 (CAF-1), cancer associated fibroblasts 2 (CAF-2), colon epithelium, immune cells (HLA genes). Scale bar 200 um. **e** Composite image with pseudo colors: colon epithelium (white), CAF-1 (cyan), CAF-2 (purple) and immune cells (yellow). Scale bar, 200 μm. **f** Zoomed-in region of the white box insert in (**e**). Scale bar, 25 μm. **g** Centroids of the segmented cell masks for CAF-1 (cyan), CAF-2 (purple), immune cells (yellow). Scale bar, 200 μm. **h** Box plots of the number of immune cells within 100 μm radius of CAF-1 (cyan) and CAF-2 (purple) cells. Immune cells were found 0.74-fold less frequently in the vicinity of CAF-2 than CAF-1. Number of cells, n: CAF-1: 2946, CAF-2: 2671, examined over 1 experiment. A technical replicate is provided as Supplementary Fig. 16. The box plots show the median (center line), the first and third quartiles (box limits), and 1.5 × the interquartile range (whiskers). $p = 1.4 \times 10^{-72}$, 2-sided Mann-Whitney U test. Source data are provided as a Source Data file.

amplification[41–44] to further enhance the signal. The development of more precise and scalable scRNA-seq methods[45,46] and the availability of more comprehensive cell atlas reference datasets should facilitate a wider array of cell types that can be mapped using FISHnCHIPs. In addition to scRNA-seq, prior information on biochemical pathway[47], transcription factor motif[48], chromatin accessibility[49], bulk gene expression[50], or even sequencing based spatial transcriptomics[51] may be used to inform the design of FISHnCHIPs gene panel.

Compared to multiplexed immunostaining methods, FISHnCHIPs offers greater flexibility and throughput, as it can exploit custom-designed but relatively inexpensive oligonucleotide probes. Besides, labeling of antibody panels often requires individual optimization, but the fluorescence signal from FISHnCHIPs is likely to be more consistent because the hybridization of probes is highly efficient across the transcriptome[15,16]. FISHnCHIPs is applicable to any cell population for which transcriptomic characteristics are known, thus allowing the interrogation of cell states not accessible by antibody-based methods. As an alternative to immunolabeling, FISHnCHIPs could be highly useful for the validation of novel cell types identified from scRNA-seq studies. Similar to multiplexed smFISH, FISHnCHIPs can be used to quantify cell types, derive zonation patterns, and analyze cell-cell interactions. However, FISHnCHIPs is not able to count the copy number of a single gene, determine its localization at the sub-cellular level or analyze signaling between specific receptors and ligands. Where it is necessary to obtain transcriptomic information at both levels, FISHnCHIPs could be combined with multiplexed smFISH to enable simultaneous cell-level and transcript-level analysis.

We foresee that the high sensitivity of FISHnCHIPs will allow the design of simpler and lower-cost spatial omics instruments, thereby improving the accessibility of spatial assays for the broader biomedical research community. Besides neuroscience and oncology, FISHnCHIPs could find broad use in other biological studies, such as mapping gene programs during embryonic development or defining multi-cellular ecosystems of infectious pathogens. We also expect FISHnCHIPs to be useful for the molecular histopathology of Formalin Fixed Paraffin Embedded tissues, where clinically actionable cell states could be diagnosed accurately and at scale.

## Methods

### Ethics statement

Experiments in this study involving human tissue were conducted in accordance with institutional ethical regulations. This study is approved by the A*STAR Research Integrity, Compliance, and Ethics Office for the use of non-individually identifiable human biological materials for in vitro research studies under application number IRB F-112.

### FISHnCHIPs gene panel design and evaluation software

Supplementary Fig. 19 summarizes the software workflow for FISHnCHIPs panel design and evaluation. To target specific cell types, FISHnCHIPs (cell-centric) either accepts user input of reference markers and cell labels or performs de novo clustering of cell types and identifies DE gene(s) as the reference marker(s). The default measure of co-expression is the Pearson's correlation coefficient. Other possible measures include mutual information, Spearman's rank correlation coefficient, and Euclidean distance[52]. To explore gene expression activities without a priori cell type clustering of the scRNA-seq data, FISHnCHIPs (gene-centric) performs either feature selection and/or dimensionality reduction (e.g., using NMF), followed by clustering analysis of the gene-gene correlation matrix to identify gene modules. In the feature gene-based method, genes that were highly correlated (> min *corr*) with a minimum number of genes (> min *genes*) were used as nodes in a network that was constructed from the gene-gene correlation matrix and partitioned using the Leiden algorithm[53]. Gene

partitions can be further sub-clustered using hierarchical clustering based on their log-transformed expression matrix. For the dimensionality reduction-based method, a non-negative matrix factorization (NMF) algorithm[31] that identifies gene programs and their relative contributions can be used. The top N genes from each program are chosen to construct the gene-gene correlation matrix. Clustering of the matrices can be refined by setting correlation ranges. We also designed a hybrid FISHnCHIPs method where the DE genes are used as features to construct the gene-gene correlation matrix to identify gene modules. We recommend FISHnCHIPs users to perform clustering in the gene-gene space to reduce crosstalk. We evaluated the output gene panel by predicting the SG and specificity, as well as by simulating the expected cell-module expression profile and clusters. We demonstrated cell-centric FISHnCHIPs for the mouse kidney library (Fig. 2), gene-centric FISHnCHIPs for the mouse cortex libraries (Figs. 3, 4), and hybrid approach for the mouse brain (Fig. 5) and human CRC library (Fig. 6). The following paragraphs describe the FISHnCHIPs panel design and evaluation process in more detail:

**Data preprocessing.** We preprocessed the scRNA-seq count matrix using the Seurat pipeline[54]. First, the QC filters empty droplets and cell doublets, i.e., cells expressing too few or too many unique genes. After QC, three versions of the gene count matrix will be prepared for different downstream analyses: (1) Scale the total counts of cells to a constant by dividing the total counts of cells and multiplying a scale factor. The cell-scaled matrix would be used for predicting the expected signal of a FISHnCHIPs panel. (2) Add a pseudo-count to the cell-scaled matrix and apply a natural log transformation. The log-transformed matrix would be used for the differential gene analysis and gene-gene correlation analysis; (3) Apply a linear transformation to the gene expression vectors, so that the mean expression of genes across cells is 0 and the variance across cells is 1. The gene-scaled matrix would be used for dimensionality reduction and heatmap visualization of the expression of individual genes.

**Panel evaluation.** A FISHnCHIPs panel can be evaluated by the SG and SSR: denoting a FISHnCHIPs panel with $n$ genes as $P_t = \{g_1, g_2, \ldots, g_i, \ldots, g_n\}$ targeting the cell type $C_t$; the number of probes for genes corresponds to $K = \{k_1, k_2, \ldots, k_i, \ldots, k_t\}$. The predicted signal of one gene $g_i$ in cell type $C_t$, denoted as $signal(g_i, C_t)$, is defined as the product of $k_i$ and the average expression of $g_i$ in cell type $C_t$. The signal of a panel $P_t$ in a cell type $C_t$, which is denoted as $signal(P_t, C_t)$, is the sum of all gene signals in the target cell type or module. Denoting $g_1$ as the reference gene, and $g_{max}$ as the gene with the maximal signal. The general SG is defined as $\frac{signal(P_t, C_t)}{signal(g_1, C_t)}$, i.e., the ratio of the panel signal to the signal of the reference gene. The conservative SG is defined as $\frac{signal(P_t, C_t)}{signal(g_{max}, C_t)}$, i.e., the ratio of the panel signal to the highest gene signal. The crosstalk can be estimated by calculating the SSR of a panel $P_t$, between cell type $C_t$ and $C_{t'}$, defined as $\frac{signal(P_t, C_{t'})}{signal(P_t, C_t)}$, i.e., the ratio of panel signal in $C_t$ to the ratio of panel signal in $C_{t'}$. The general signal specificity is defined as the ratio of the panel signal in the target cell type to the panel signal in all off-target cell types. The conservative signal specificity is defined as the ratio of the panel signal in the target cell type to the panel signal in the cell cluster with the highest predicted crosstalk. We used the general SG for the cell-centric mouse kidney panel and the conservative SG for all other FISHnCHIPs panels. A FISHnCHIPs panel can be further evaluated by re-clustering the scRNA-seq dataset using the module-cell expression matrix. The module-cell expression matrix is calculated from the cell-scaled expression matrix, by taking the sum of cell counts of genes in the same group. Considering the module as a meta-gene, the module-expression matrix can be taken as a meta-gene expression matrix. Consequently, conventional clustering methods used to process

single-cell gene count matrices can be applied. A module-cell expression heatmap and dimensionality reduction visualization tools (such as UMAP or tSNE) could be used to simulate the reconstruction of cell types from a FISHnCHIPs assay.

**Mouse kidney panel (Fig. 2)**
The scRNA-seq data and cell labels of the mouse kidney were retrieved from NCBI Gene Expression Omnibus (GEO) under accession GSE115746[27]. Genes with the highest log fold-change of the average expression between the targeting clusters and other clusters were selected as reference markers. Cells with <200 or >3000 unique expressed genes were removed. Cells with mitochondrial genes >50% were removed. Genes that were expressed in <10 cells were removed. Cells were then scaled to a sequence depth of 10,000 per cell and log-transformed with a pseudo-count of 1. Genes were scaled so that the mean expression across cells was 0 and the variance across cells was 1. For each cluster, genes correlated to the reference markers and with Pearson Correlation >0.5 were selected. If there were <15 genes highly correlated with the reference, the top 15 genes were selected. For all clusters, we removed genes that appeared more than once. For glomerular endothelial cells, the top maker *Plat* was only expressed in 59.5% of glomerular endothelial cells, and it was also highly expressed in glomerular podocytes. Therefore, the *Emcn* was used as the reference marker instead of *Plat*. For renal macrophages, both *C1qa* and *C1qb* were used as references. We chose 5 cell types for imaging (Fig. 2), but computationally evaluated all the previously annotated cell types (Supplementary Fig. 2).

**Feature gene-based mouse cortex panel (Fig. 3)**
A scRNA-seq dataset[29] of the mouse primary visual cortex (VISp) was used for the mouse brain panel design. First, the cells were scaled to 10,000, then the gene expression in cells was binarized by the mean expression of all genes across all cells. Genes that were expressed in <5 cells or >80% of the total number of cells were filtered out. Gene names starting with "Mt" or "Gm" followed by digits were removed. 330 genes highly correlated to at least 5 genes with a correlation >0.7 were selected as candidates. A graph was created from the 330 by 330 correlation matrix, removing edges with low correlation (<0.6). Leiden partitioning on the graph with 330 candidate genes generated 11 clusters. We further performed hierarchical clustering on the Leiden clusters based on gene expression, cutting the dendrogram of genes into k subclusters: $k = 6$ for big clusters (>30 genes); $k = 4$ for mid-size clusters (11-30 genes); $k = 2$ for small clusters (6–10 genes); $k = 1$ for very small clusters (<6 genes). There were 255 genes distributed in 18 modules after removing subclusters with single genes, genes not found in our probe design transcriptome database (*Hsp25-ps1* and *Gstm2-ps1*) or associated with multiple IDs in our probe design transcriptome database (*Schip1*). Functional enrichment analysis, known as gene set enrichment analysis, on the panel genes was performed using g:GOSt[55].

**Dimensionality reduction-based mouse cortex panel (Fig. 4)**
Non-negative matrix factorization (NMF) provides a low rank approximation of the gene cell matrix by a product of two non-negative matrices, and is able to capture the structures of coordinated gene expression in scRNA-seq data[31,56]. The gene-contribution matrix of the mouse visual cortex neurons was downloaded from[31]. The highest contributing 50 genes were selected from the 20 factors. Gene names starting with the "Gm" followed by digits were removed. Clustering of the gene-gene correlation matrices resulted in one or more gene modules per program. By comparing the gene expression heatmap and the gene-gene correlation matrices, we noticed that most genes with Pearson's correlation (r) higher than 0.3 showed expression that spanned multiple programs and were markers associated with the major cell types (such as for all inhibitory neurons). Therefore, we removed genes with r higher than 0.3 and lower than 0.02. There were

311 genes distributed in 20 programs after further discarding genes with no probes found.

## 674-gene mouse brain panel (Fig. 5)

Utilizing the subcluster labels provided by the mouse brain Drop-seq scRNA dataset[34], we identified a maximum of 50 DE genes with at least 0.25-fold difference for all subclusters, employing the Wilcoxon Rank Sum test algorithm implemented in Seurat. For each subcluster, genes with the lowest correlation to any DE gene were removed until the minimal Pearson correlation matrix of the remaining genes was greater than 0.1. To further refine the quality of the panel, genes starting with "mt" and small modules with fewer than 5 genes were excluded, resulting in 53 gene modules containing 674 genes. To evaluate the panel, we re-clustered the scRNA-seq dataset using the 53 modules as features and calculated the Adjusted Rand Index using the "aricode" package in R. To provide further comparisons, we also simulated single gene-based multiplexed FISH assays by re-clustering the scRNA-seq data using 1000, 2000, and 3000 highly variable genes as features detected by the "vst' method provided in the Seurat package (Supplementary Fig. 9b).

## Human colorectal cancer panel (Fig. 6)

We previously identified two cancer-associated fibroblasts (CAFs) subtypes using scRNA-seq[37]. We further confirmed these two subtypes using a more recent scRNA sequencing dataset[57] (Supplementary Fig. 15). Genes that were expressed in <5 cells or >70% of the total number of cells were filtered out. Gene names starting with "Rp", "Mt" or "Gm" followed by digits were removed. Based on the 125 selected marker genes, a graph was created from the gene-gene correlation matrix, removing edges with low correlation (<0.7). Leiden partitioning on the graph yielded ~20 modules and we selected 4 modules highly expressed in the two CAFs, epithelial, and immune cells for demonstrating FISHnCHIPs.

## FISHnCHIPs library design and probe sequences

For all the genes, 25-nucleotide target regions were identified using a previously published algorithm[58]. Briefly, reference transcript sequences were downloaded from the GENCODE website (human v24 and mouse m4). A specificity table was calculated using 15-nucleotide seed and 0.6 specificity cutoff was used. Quartet repeats ("AAAA", "TTTT", "GGGG", and "CCCC") were excluded from the possible target regions. The list of FISHnCHIPs and readout probe sequences can be found in Supplementary Data 1. We generated 56 readout probe sequences initially, but B16, B48 and B55 were not used.

## Probe amplification and preparation

The oligonucleotide pools (Genscript) were amplified enzymatically to generate the encoding probes used for sample staining[8]. First, the oligonucleotide pool was amplified by limited-cycle PCR using Phusion Hot Start Flex 2× Master Mix, with an annealing temperature of 68 °C. The T7 promoter sequence was introduced on the reverse primer during PCR. Further amplification was achieved by in-vitro transcription that was performed overnight using a high-yield in vitro transcription kit (NEB, cat. no. E2050S). Reverse transcription was then performed on the RNA template using Maxima H- Reverse Transcriptase (Thermo Fisher, cat. no. EP0753) to create a DNA-RNA hybrid. The RNA part was then cleaved off with alkaline hydrolysis, leaving behind a single-stranded DNA (ssDNA) which was then purified via magnetic bead purification and eluted in nuclease-free water (Ambion, cat. no. AM9930). The primers used for PCR are as follows:

Mouse Kidney Library for Fig. 2:
Forward primer: 5'-CTATGCGCTATCCCGGACGC-3'
Reverse primer: 5'-TAATACGACTCACTATAGGGTCGCATATCCG TACCGGC-3'.
Mouse Cortex Library for Fig. 3:

Forward primer: 5'-CCGTTCAAGACTGCCGTGCTA-3'
Reverse Primer: 5'-TAATACGACTCACTATAGGGCTAGGGAGCCT ACAGGCTGC-3'
Mouse Cortex Library for Fig. 4:
Forward primer: 5'- TTGCGTTCGGTCTGAATGCG-3'
Reverse Primer: 5'- TAATACGACTCACTATAGGGACTCCTGCTCT TTGGGTCCG-3'
Mouse Brain Library for Fig. 5:
Forward primer: 5'-CGCCCTAATCTCCGCTTGGG'-3'
Reverse Primer: 5'-TAATACGACTCACTATAGGGGCTTCGACCGAG GGCGAAAT'-3'
Human Colorectal Cancer Library for Fig. 6:
Forward primer: 5'- TGCCCGCCTTTCGTTACTCA -3'
Reverse Primer: 5'- TAATACGACTCACTATAGGGCGCAATCGT CGGCTAACGGT -3'.

## Coverslip functionalization

Coverslips were coated before being used for tissue sectioning[21,59]. The coverslips (Warner Instruments, cat. no. 64–1500) were cleaned by gently shaking in 1 M KOH for 1 h and rinsed thrice with Milli-Q water. The coverslips were rinsed with 100% methanol, then immersed in an amino-silane solution (3% vol/vol (3-aminopropyl) triethoxysilane (Merck cat no. 440140), 5% vol/vol acetic acid (Sigma, cat. no. 537020) in methanol) for 2 min at room temperature before being rinsed three times with Milli-Q water and dried in an oven at 47 °C overnight. Functionalized coverslips were then used immediately or stored in a dry, desiccated environment at room temperature for several weeks.

## Mouse tissue sample preparation

8-week-old C57BL/6nTAc female mice, purchased from InVivos, were used for all animal experiments in this study. All animal care and experiments were carried out in accordance with Agency for Science, Technology and Research (A*STAR) Institutional Animal Care and Use Committee (IACUC) guidelines (IACUC #211580). The mice were euthanized on the day of arrival, and their kidneys and brains were quickly collected and frozen immediately in optimal cutting temperature compound (Tissue-Tek O.C.T. Compound; Tissue-Tek, SKU #4583), before storing at −80 °C. The fresh frozen samples were then sectioned with a cryostat into 7 μm sections directly onto functionalized coverslips. For the comparison between 10× and 60× objectives (Supplementary Fig. 13), adjacent mouse sagittal brain sections were used. Sections were air-dried for 5 min at room temperature before being fixed with 4% vol/vol paraformaldehyde in 1 × PBS for 15 min. Following fixation, samples were rinsed once with 1 × PBS and were either permeabilized immediately in 0.5% TritonX-100 in 1 × PBS for 10 min at room temperature, or permeabilized in 70% ethanol overnight at 4 °C, or stored at −80 °C. No sample-size estimate was performed, since the goal was to demonstrate a technology.

## Human colorectal cancer tissue sample preparation

To demonstrate the FISHnCHIPs technology, an aliquot from a non-individually identifiable tumor colon tissue was collected and frozen on dry ice immediately after resection and stored at −80 °C. Prior to sectioning, tissue was embedded in optimal cutting temperature compound. Sections were obtained as described above, and following fixation, samples were rinsed once with 1 × PBS before being permeabilized immediately in 70% ethanol overnight at 4 °C. Sections were further permeabilized in 0.5% TritonX-100 in 1 × PBS at room temperature for 15 min.

## FISHnCHIPs and smFISH staining

After permeabilization, the tissue sample was rinsed thrice with 1 × PBS, followed by a rinse with 2 × SSC. The encoding probes were diluted in a 20% or 30% hybridization buffer to a final concentration of 1–2 nM per probe. The 20% hybridization buffer composed of 20%

deionized formamide (Ambion™ Cat: AM9342) (vol/vol), 1 mg ml-1 yeast tRNA (Life Technologies, cat. no. 15401-011) and 10% dextran sulfate (Sigma, cat. no. D8906) (wt/vol) in 2× SSC. The sample was stained with the encoding probes for 16 to 48 h at 37 °C or 47 °C. Following hybridization, the sample was washed in a 20% formamide wash buffer, containing 20% deionized formamide and 2 × SSC, twice, incubating for 15-30 min at 37 °C or 47 °C per wash. The wash buffer was then removed, and the sample was washed twice with 2 × SSC. The staining and washing conditions were optimized individually for each sample type. DAPI (Sigma, cat. no. D9564) was stained at a concentration of 1 µg/ml in 2 × SSC for 10 min at room temperature. The sample was then washed thrice with 2x SSC and were either imaged immediately or stored at 4 °C in 2 × SSC for no longer than 12 h before imaging. For smFISH of *DCN*, *MMP2*, *TAGLN*, *ACTA2*, and *SPARC* (Stellaris RNA FISH probe sets, LCG Biosearch Technologies), the probes were diluted with 10% hybridization buffer, and samples stained overnight at 37 °C. Samples were then washed twice with a 10% formamide wash buffer for 15 min at 37 °C per wash, before rinsing with 2 × SSC and subsequent imaging. Sequences of the smFISH probes can be found in Supplementary Data 5.

## FISHnCHIPs imaging cycle

A flow chamber (Bioptechs, cat. no. FCS2) that could be secured to the microscope stage was used to mount the sample. Readout probe hybridization was performed directly in the flow chamber by buffer exchange that was controlled by a custom-built, computer-controlled fluidics system[8]. All the buffer solutions (-1 ml per exchange) were flowed within 1 min. 10 nM of fluorescently labeled readout probe in 10% high-salt hybridization buffer was flowed into the chamber and incubated for 10 min at room temperature. The 10% high-salt hybridization buffer was composed of 10% deionized formamide (vol/vol) and 10% dextran sulfate (Abcam, cat. no. ab146569/Sigma, cat. no. D8906) (wt/vol) in 4 × SSC. Following hybridization, the sample was rinsed with 2 × SSC before flowing in 10% formamide wash buffer containing 0.1% TritonX-100. 2 × SSC was flowed once more before imaging buffer. The imaging buffer consisted of 2 × SSC, 10% glucose, 50 mM Tris-HCl pH 8, 2 mM Trolox (Sigma, cat. no. 238813), 0.5 mg/ml glucose oxidase (Sigma, cat. no. G2133) and 40 µg/ml catalase (Sigma, cat. no. C30). To remove the fluorescent signals, the samples were washed with 55% formamide wash buffer containing 0.1% TritonX-100. This hybridization and wash cycle were repeated until all the readout probes were imaged.

## Imaging set-up 1

FISHnCHIPs imaging was performed on a home-made microscope and fluidics set-up[21]. Briefly, the microscope was constructed around a Nikon Ti2-E body, Marzhauser SCANplus IM 130 mm × 85 mm motorized X–Y stage, a Nikon CFI Plan Apo Lambda 60× 1.4-n.a. oil-immersion objective, and an Andor Sona 4.2B-11 sCMOS camera. For the whole slide imaging experiment (Fig. 6), the Nikon CFI Plan Apo 10× 0.5-n.a. water-immersion objective was used. The DAPI channel was excited by a Coherent Obis 405 100-mW laser. MPB Communications fiber lasers were used as illumination for Alexa594 (592 nm), Cy5 (647 nm) and IRDye 800CW (750 nm), respectively: 2RU-VFL-P-500-592-B1R (500 mW), 2RU-VFL-P-1000-647-B1R (1000 mW) and 2RU-VFL-P-500-750-B1R (500 mW). The Nikon Perfect Focus system was used to maintain focus while imaging, and in each imaging cycle, one Z position was imaged for each field of view. The Perfect Focus system was not used when imaging under the 10× water-immersion objective. Images were acquired at different exposure times (1 s, 500 ms, and 1 s with 60× and 3 s, 3 s, and 5 s with 10× for Alexa594, Cy5, and IRDye 800CW respectively) to avoid saturating the camera.

## Imaging set-up 2

A custom-built microscope constructed around a Nikon Ti2-E body, Marzhauser SCANplus IM 130 mm × 85 mm motorized X–Y stage, and a

pco.edge 4.2 BI-USB Back Illuminated sCMOS camera was used. A custom, fiber-coupled laser box from CNI laser was used as illumination for DAPI (405 nm), Alexa Fluor 488 (488 nm), Alexa Fluor 594 (588 nm), Cy5 (637 nm) and IRDye 800CW (750 nm). Custom multi-wavelength filters, 445/503/560/615/683/813 (Semrock) and 405/473/532/588/637/730 (Semrock), were used. The following objectives were tested: Nikon CFI Plan Apo Lambda 10× 0.45-n.a. air objective (MRD00105), Nikon CFI Plan Apo 10× 0.5-n.a. water-immersion objective (MRD71120), Nikon CFI Plan Fluor 20× 0.75-n.a. water-immersion objective (MRH07241), Nikon CFI S Plan Fluor ELWD 20× 0.45-n.a. air objective (MRH08230), Nikon CFI Apo LWD Lambda S 40× 1.15-n.a. water-immersion objective (MRD77410), and Nikon CFI Plan Apo Lambda 60× 1.4-n.a. oil-immersion objective (MRD01605). At 40× and 60×, the focus was maintained using the Nikon Perfect Focus system. One Z position was imaged per field of view. This set up is used for objective lenses comparison experiment (Supplementary Fig. 8) and for immunofluorescence imaging (Supplementary Fig. 17).

## Immunofluorescence staining

Tissues were rinsed with 1 × PBS thrice at room temperature. Blocking was done with 1% BSA (NEB) and 0.1% Tween-20 in 1 × PBS for 1 h at room temperature. Tissues were stained at 4 °C overnight using the following antibodies diluted in blocking solution: anti-LUM (Abcam, ab168384; clone EPR11380(B); Lot GR121948-4; 1:75), anti-MMP2 (Abcam, ab97779; Lot GR3448382-1; 1:200), anti-α-SMA (Abcam, ab7817; clone 1A4; Lot 1009584-11;1:600), and anti-PDGFA (Santa Cruz Biotechnology, sc-9974; clone E-10; Lot C0222; 1:600). PDPN was detected using AF488-conjugated primary antibody (BioLegend, 337005; clone NC-08; Lot B360564; 1:75). Secondary antibody staining was then carried out for 1 h at room temperate using anti-mouse AF594 (ThermoFisher, A11005; Lot 2538976; 1:1000) and anti-rabbit AF488 (ThermoFisher, A11008; Lot 2557379; 1:1000). Finally, samples were stained with anti-CD68 (Cell Signaling Technology, #79594; clone D4B9C; Lot 779594S; 1:50) overnight at 4 °C. After washing with 1 × PBS three times, tissues were counterstained with DAPI (Sigma) before mounting (Vectashield, H-1700-10).

## FISHnCHIPs image processing and data analysis software

A custom pipeline (Supplementary Fig. 4) was created to align the images (DAPI images, FISHnCHIPs images, and background images), segment, and cluster cell types. First, nuclei masks were obtained by performing nucleus segmentation using the deep learning based Cellpose algorithm[60] or the watershed algorithm. FISHnCHIP images were registered to the DAPI image by phase correlation using a subpixel registration algorithm provided in the Scikit-Image package[61]. Subsequently, background images (after the 55% formamide wash, images were taken and used to estimate tissue autofluorescence background) were subtracted from the FISHnCHIPs images after alignment (i.e., applying the same shifts). The nuclei masks obtained from the segmentation of DAPI were dilated to create cell masks, which were applied to all background subtracted FISHnCHIPs images. A FISHnCHIPs intensity matrix was constructed for cell type clustering and subsequent analyses. The intensity matrix was clustered using the Louvain algorithm after QC and normalization. Cell clusters were visualized in a heatmap, dimensionality reduction plot, as well as a cluster map.

## Gain and crosstalk analysis for mouse kidney (Fig. 2)

The nuclei segmentation and image alignment were performed as described above. Nuclei masks smaller than 3000 pixels were discarded. Nuclei masks were dilated by 5 pixels for creating cell masks. Images were normalized by dividing by the 99th percentile of pixel intensities. A cell-by-channel-intensity matrix was constructed by calculating the mean fluorescence intensity per cell using the cell masks. Since we chose to image only five kidney cell types in this experiment,

cells with normalized intensity lower than 0.5 were dropped (keeping only ~18.6% of the cells that were brightly labeled by FISHnCHIPs). Qualified cells with the highest normalized intensity across the channels were assigned to be the corresponding cell type. The FISHnCHIPs fluorescence SG was calculated by taking the ratio of the mean FISHnCHIPs intensity to the mean smFISH intensity in the same cell (the same cell masks were applied to both FISHnCHIPs and smFISH images as they were imaged sequentially on the same sample). The crosstalk of FISHnCHIPs was estimated by calculating the Mander's overlap coefficient[62], a metric that quantifies the degree of co-localization of objects in a pair of images (and was originally developed for dual-color confocal microscopy). It is the fraction of overlap between two channels: $M_1 = \frac{\sum (C_1 > t_1) \& (C_2 > t_2)}{\sum (C_1 > t_1)}; M_2 = \frac{\sum (C_1 > t_1) \& (C_2 > t_2)}{\sum (C_2 > t_2)}$, where $t_1$ and $t_2$ were the thresholds for binarizing the two channels $C_1$ and $C_2$ respectively.

### Figure 3 18-module mouse cortex data analysis

The nuclei segmentation and image alignment were performed as described above. Nuclei masks smaller than 3000 pixels were discarded. Nuclei masks were dilated by 15 pixels for creating cell masks. Images were normalized to their 99th percentile of pixel intensities. The cell-by-module-intensity matrix was constructed by taking the mean intensity of the segmented cell masks. Cells with total intensity lower than the 15th percentile were removed for QC, resulting in 72.04% nuclei passing QC. The cell-by-module-intensity matrix was used for clustering using the Seurat package. Modules were z-scaled before calculating principal components and dimensionality reduction projection. Clustering analysis was performed using the Louvain clustering algorithm. Cells were clustered at a resolution of 0.8 using the top 10 PCs with 20 nearest neighbors. Finally, the cell clusters were mapped back to the location of cell masks to reconstruct the spatial map.

### Figure 4 mouse cortex neuronal subtypes data analysis

The nuclei segmentation and image alignment were performed as described above. Nuclei masks smaller than 3000 pixels were discarded. Nuclei masks were dilated by 10 pixels for creating cell masks. Images were normalized to their 99th percentile of pixel intensities. The cell-by-program-intensity matrix was constructed by taking the mean intensity of cell masks. Images were cropped to contain only the cortical region as shown in Fig. 4. Cells with total intensity lower than the 20th percentile were removed for QC. The clustering analysis was performed as described above but at a higher resolution of 1.2. 5 out of 18 clusters (29.7% of the cells) contained cells with weak or no neuronal expression signature, which were then removed. As a result, 50.3% of all cells (defined by DAPI) were qualified as neurons. To quantify the cortical depth of neuron cells, edges from two circles with the same radius $R = 25,500$ pixels were used to cover the regions with excitatory neurons as shown in Fig. 4. The distance between the two centers was 10,000 pixels. The normalized depth of cells was defined as the distance to the outer edge divided by the distance between the two centers. The cortical depth cell intensity heatmap was plotted by arranging cells with increasing depth (Supplementary Fig. 7). The cell density along the cortical depth was estimated by applying a kernel density estimate with a 0.05 Gaussian kernel.

### Figure 5. 53-module large FOV mouse brain data analysis

To generate the cell-by-module intensity matrix and cell positions, we normalized the nuclei images to the 99th percentile of pixel intensities and utilized the same nuclei segmentation pipeline as mentioned above. Each FISHnCHIPs image was registered to their corresponding DAPI images, and the shifts were recorded. Shifts exceeding 50 pixels in any direction were discarded. The average shifts were then applied to all fields of view. To correct illumination variations between fields of view, we subtracted the 60th percentile intensity of pixels outside the

cell masks. Cells with low intensity (<0.2%) across all modules, or with high intensity (>98%) across over 30 modules were removed. We initially constructed a graph of cells based on 15 nearest neighbors using the top 20 PCs and performed Leiden clustering at a resolution of 2. 133 cells (0.25%) from 2 of the preliminary clusters were affected by the autofluorescence of a dust particle in the sample and were dropped from further analysis. 54,834 (97.3%) qualified cells were clustered with a lower resolution of 0.6, resulting in 18 clusters or cell types. The blood vessel associated cells cluster and the inhibitory neurons cluster showed finer structure in the UMAP and were further sub-clustered. To verify the cluster annotations, we performed integration analysis using the Harmony algorithm[63] between FISHnCHIPs and scRNA-seq (Supplementary Fig. 11). To ensure compatibility, we cropped the FISHnCHIPs data to the frontal cortex region. Additionally, we subsampled the scRNA-seq data randomly to balance the number of cells, following the recommendation by the Harmony authors. Normalization and scaling were applied to both scRNA-seq and FISHnCHIPs data before integration. We were unable to annotate one of the clusters (2773 or 5% of the cells), as they exhibit low level expression across both the neuronal and non-neuronal modules and are spatially heterogeneous. From the integration analysis[63], we observed that these cells are in close proximity to the polydendrocytes and excitatory neuron clusters. Based on this observation, the "Unknown" cluster may represent one or multiple genuine cell populations that was not resolved by our current probe set.

### Proximity of cancer-associated fibroblasts (CAFs) to immune cells in human colorectal cancer tissue (Fig. 6)

The fibroblasts and immune cells were segmented using the watershed segmentation algorithm provided in the Scikit-image package[61]. The cutoff threshold and opening threshold for watershed segmentation were adjusted manually for each cell type. Using the centroids of the segmented cell masks, we calculated the number of immune cells within a 100 μm radius of CAF-1 or CAF-2 cells. We found that there were significantly greater numbers of immune cells closer to CAF-1 cells compared to CAF-2 cells (2-sided Mann-Whitney U test). This result was consistent with a visual inspection of cell positions (Fig. 6 and Supplementary Fig. 16).

### Statistics and reproducibility

FISHnCHIPs was demonstrated in multiple tissue types, indicating that the method is robust and reproducible. All results shown are from $n = 1$ experiment. Figures 2, 3, 4, 5, and 6 experiments were repeated at least once independently with similar results. All Supplementary Figs. experiments (except for Supplementary Figs. 8, 13, 17, and 18) were repeated at least once independently with similar results.

### Reporting summary

Further information on research design is available in the Nature Portfolio Reporting Summary linked to this article.

## Data availability

Source data are provided with this paper. The mouse kidney scRNA-seq dataset used in this study is available in the NCBI GEO database under accession code GSE107585[27]. The mouse brain scRNA-seq datasets used in this study are available in the NCBI GEO database under accession code GSE115746 and the Dropviz website [http://dropviz.org/][29,34]. The human colorectal cancer scRNA-seq datasets used in this study are available in the NCBI GEO database under GSE81861 and GSE178341[37,57]. The raw image files are available from the corresponding authors (chenck@gis.a-star.edu.sg; nigel_chou@gis.a-star.edu.sg). We prefer to share the dataset electronically and are committed to responding to such requests within a week upon reasonable request. They can be used for academic and non-commercial purposes. Source data are provided with this paper.

## Code availability

The software to design FISHnCHIPs gene panel and analyze FISHnCHIPs data is available at the following repository: https://github.com/KHChenLab/FISHnCHIPs. It is also available at Zenodo: https://doi.org/10.5281/zenodo.10146111.

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

## Acknowledgements

We thank Tim Stuart, Fong Kuan Wong, Winston Zhao, Vipul Singhal, Jinyue Liu, Maurice Lee, Hwee Kuan Lee, and Antonia Monteiro for insightful discussions, technical assistance, and comments during the preparation of the manuscript. We thank Iain Tan, Anna Gan, Polly Poon, and team from the National Cancer Center Singapore for the non-identifiable CRC tissue aliquots used in this study. This work is supported by National Medical Research Council of Singapore grant OFIRG20nov-0056 (K.H.C.) and National Research Foundation grant NRF-CRP25-2020-0001 (K.H.C.); Agency for Science, Technology, and Research grant IAF-PP-H18/01/a0/020 (S.P.); Agency for Science, Technology, and Research grant CDF 202D800010 (N.C.); N.C. is a recipient of the GIS fellowship support from the Genome Institute of Singapore.

## Author contributions

Conceptualization: K.H.C.; Data Curation: K.H.C., W.Y.S., X.Z.; Formal Analysis: X.Z., N.C., K.H.C.; Funding Acquisition: K.H.C., N.C., S.P.; Investigation: W.Y.S., N.H., J.J.L.G., L.J.; Methodology: K.H.C., X.Z., S.P.; Resources: K.H.C., S.P.; Software: X.Z., N.C., T.H.C., J.B.; Supervision: K.H.C., N.C., S.P.; Validation: N.H., L.J.; Visualization: K.H.C., N.C., W.Y.S., X.Z.; Writing—original draft: K.H.C., N.C., X.Z., W.Y.S.; Writing—review & editing: K.H.C., N.C., X.Z., W.Y.S.

## Competing interests

The authors declare the following competing financial interests: The FISHnCHIPs technology described in the manuscript was filed under Singapore Patent Application No. 0202260245V on 29 Nov 22. We are in the process of filing an international PCT. Agency for Science Technology and Research (A*STAR) is the patent applicant and the inventors are K.HC., X.Z., W.Y.S., N.H., J.B., N.C. and J.J.L.G. The remaining authors declare no competing interest.
