## [Peer Review File · Nature Communications]

Highly sensitive spatial transcriptomics using FISHnCHIPs of multiple co-expressed genesREVIEWER COMMENTS

Reviewer #1 (Remarks to the Author):

The revised manuscript addressed many important technical concerns, including more validations for the cluster assignments of mouse brain tissues and validation experiments of CRC tissues. I would also like to thank the authors for their detailed clarifications regarding the key advantage of FISHnCHIPs compared to other spatial transcriptomic methods. Overall, I think the paper is much improved and the method will be of interest to the field. I only have two main concerns at this point:

1. Although the paper has sufficiently demonstrated its advantage in fast cell typing with higher sensitivity, it seems to show less cell type diversity in the sagittal section. This is certainly reflected when comparing scRNA-seq and FISHnCHIPs annotation in reviewer table 1. Maybe this is due to the limited modules selected for data collection.
2. Another concern is the lack of noteworthy biological discoveries in the paper. I have asked this point in my previous comments, but I see that the authors have highlighted the novelty of the technology instead. I think this should not distract from the missing biological insights. From the technology point of view, I think it's suitable for Nature Communications. I will leave the editor to decide.

Reviewer #2 (Remarks to the Author):

The authors have addressed most of my comments, and I commend the authors for the thoroughness of the responses.

Two items:

- 1) Could the authors include reviewer table 2 and associated discussion of it into the text of the manuscript.
- 2) I understand for the single-cell UMAPs the authors have computationally clustered it via marker genes, and/or HVG1000 in figures (ext 4c, 3c, and 9). Can the authors please include ground truth clustering of the single-cell using all the data (using default variable gene identification via mean/variance) alongside for reference.

Reviewer #1 (Remarks to the Author):

The revised manuscript addressed many important technical concerns, including more validations for the cluster assignments of mouse brain tissues and validation experiments of CRC tissues. I would also like to thank the authors for their detailed clarifications regarding the key advantage of FISHnCHIPs compared to other spatial transcriptomic methods. Overall, I think the paper is much improved and the method will be of interest to the field. I only have two main concerns at this point:

1. Although the paper has sufficiently demonstrated its advantage in fast cell typing with higher sensitivity, it seems to show less cell type diversity in the sagittal section. This is certainly reflected when comparing scRNA-seq and FISHnCHIPs annotation in reviewer table 1. Maybe this is due to the limited modules selected for data collection.

We acknowledge that the key strength of FISHnCHIPs is to map major cell types with a large number of co-expressed genes. Nonetheless, we showed that FISHnCHIPs was still able to image many cell subtypes, such as the various subtypes of inhibitory neurons. In future work, we plan to further expand the number of modules by developing a combinatorial barcoding strategy to enable an even more comprehensive profiling of cell type diversity.

2. Another concern is the lack of noteworthy biological discoveries in the paper. I have asked this point in my previous comments, but I see that the authors have highlighted the novelty of the technology instead. I think this should not distract from the missing biological insights. From the technology point of view, I think it's suitable for Nature Communications. I will leave the editor to decide.

We acknowledge that there is indeed a lack of noteworthy biological discoveries in the paper. We wanted to introduce the scientific community to a new spatial technique, which is more cost-effective and faster for mapping cell types. Because of the much higher cellular throughput afforded by FISHnCHIPs, we also anticipate that it can facilitate the imaging of rare cell types in tissues, and thus we have added to the discussion another potential area for biological discoveries.

Page 9, line 23; "The high-throughput, large field-of-view profiling enabled by FISHnCHIPs could facilitate detection of rare cell populations within complex tissues. For instance, stem cell niches housing slowly dividing stem cells often occur at low frequencies interspersed throughout a tissue. By performing rapid whole-tissue scans, FISHnCHIPs can help identify these niche locations based on their expression signatures. Linking rare cell gene expression patterns to their surrounding microenvironments could resolve external events or cell-cell interactions that influence stem cell fates. This is especially important for revealing the potentially pathologic roles of certain tumour microenvironments in seeding cancer stem cells."

Reviewer #2 (Remarks to the Author):

The authors have addressed most of my comments, and I commend the authors for the thoroughness of the responses.

Two items:

1) Could the authors include reviewer table 2 and associated discussion of it into the text of the manuscript.

We thank the reviewer for this suggestion. We have added this discussion in the main text:

Page 8, line 13; “Notably, we observed that a much higher proportion of cells (97%) passed quality control in this 53-module library compared to previous libraries (Extended Data Fig. 14). This indicates that if FISHnCHIPs is designed to target a broad range of cell types, almost all cells imaged can be profiled, further demonstrating the high detection efficiency and sensitivity of FISHnCHIPs.”

We have also added the new Extended Data Figure 14, which is the reviewer table 2.

2) I understand for the single-cell UMAPs the authors have computationally clustered it via marker genes, and/or HVG1000 in figures (ext 4c, 3c, and 9). Can the authors please include ground truth clustering of the single-cell using all the data (using default variable gene identification via mean/variance) alongside for reference.

We thank the reviewer for this suggestion, and have now performed ground truth clustering of the single-cell data using default variable gene identification (Seurat). We have added this simulation result (2000 highly variable genes, the default setting for feature selection in Seurat) as Extended Data Figure 9c. We have also clarified the software package and parameter settings used in the methods:

Page 34, line 19; “To provide further comparisons, we also simulated single gene-based multiplexed FISH assays by re-clustering the scRNA-seq data using 1000, 2000, and 3000 highly variable genes as features detected by the ‘vst’ method provided in the Seurat package (Extended Data Fig. 9b).”

Extended Data Figure 9c;

Note that the ARI score for the HVG2000 simulation at a resolution of 0.1 is 0.568, which is lower than the ARI scores for the HVG1000 and the FISHnCHIPs 53-module.

Extended Data Fig. 9 (b-e) UMAP representation for cells in the scRNA-seq dataset predicted from different feature sets, namely b) 1,000 highly variable genes; c) 2,000 highly variable genes (Seurat default); d) 3,000 highly variable genes; e) 53 modules presented in Fig.5; **(f)** the ARI of clustering cells at a resolution of 0.1 using b-e as features against the labels from the scRNA-seq dataset as ground truth.

REVIEWERS' COMMENTS

Reviewer #1 (Remarks to the Author):

The authors have addressed my comments. I have no additional critiques.

Reviewer #2 (Remarks to the Author):

The authors have addressed all of my comments.